# Walking Step Monitoring with a Millimeter-Wave Radar in Real-Life Environment for Disease and Fall Prevention for the Elderly

**DOI:** 10.3390/s22249901

**Published:** 2022-12-16

**Authors:** Xuezhi Zeng, Halldór Stefán Laxdal Báruson, Alexander Sundvall

**Affiliations:** Department of Electrical Engineering, Chalmers University of Technology, SE412-96 Gothenburg, Sweden

**Keywords:** radar, gait analysis, millimeter wave, micro-Doppler, FMCW, CW, fall prevention, illness prediction

## Abstract

We studied the use of a millimeter-wave frequency-modulated continuous wave radar for gait analysis in a real-life environment, with a focus on the measurement of the step time. A method was developed for the successful extraction of gait patterns for different test cases. The quantitative investigation carried out in a lab corridor showed the excellent reliability of the proposed method for the step time measurement, with an average accuracy of 96%. In addition, a comparison test between the millimeter-wave radar and a continuous-wave radar working at 2.45 GHz was performed, and the results suggest that the millimeter-wave radar is more capable of capturing instantaneous gait features, which enables the timely detection of small gait changes appearing at the early stage of cognitive disorders.

## 1. Introduction

Improvements in public health have led to a significant increase in life expectancy, with the consequence of an increasingly aging population. According to WHO, the proportion of the world’s population over 60 years will double from approximately 11% to 22% between 2000 and 2050. The absolute number of people aged 60 years and over is expected to increase from 605 million to 2 billion over the same period [1].

The normal aging process entails declines in both cognitive and physical functions [2] that largely affect the quality of life for the elderly. It has long been known that there is a direct relationship between cognitive impairment severity and increased gait abnormalities [3]. Early motor dysfunction co-exists with or even precedes the onset of cognitive decline in older adults [4]. For example, gait patterns tend to differ from their normal behavior at the early onset of some neurodegenerative diseases, such as Alzheimer’s and Parkinson’s. A person in the primary phase of Parkinson’s tends to make small and shuffled steps and may also experience difficulties in performing key walking events, such as starting, stopping, and turning [5]. Short shuffling steps with difficulty lifting the feet off the ground were reported to be associated with an increased risk of developing dementia [6].

Gait impairments are also associated with fall incidents [7], which are considered to be a major risk for the elderly living independently, as falls often result in serious physical and psychological consequences, or even death. The rapid detection of fall incidents can reduce the mortality rate and raise the chance of surviving the event, but predicting the fall risk and preventing fall occurrence is of the uttermost importance. Several studies have identified gait abnormalities as predictors of fall risk [8,9,10]. However, the gait changes that appear at the early development stage of neurodegenerative diseases or are associated with fall risk are usually too subtle and discrete to be detected by clinical observation alone. Objective, quantitative, and continuous measurements and assessments of gaits are needed in order to detect these clinically relevant gait changes, enabling the timely introduction of individually tailored interventions.

Rapid progress in new technologies has given rise to devices and techniques that allow for an objective measurement of different gait parameters, resulting in a more efficient measurement and providing specialists with a large amount of reliable information on patients’ gaits. These technologies can be broadly divided into wearable and non-wearable methods. The wearable methods [11] require the person to wear sensors in specific locations on their body, which can be incorrectly applied or forgotten completely, especially by the elderly. In addition, the sensors may obstruct natural movement and affect the gait. For these reasons, wearable methods are not applicable for long-term monitoring purposes. Non-wearable gait analysis methods usually involve the use of motion capture cameras or floor sensors [12]. They are usually laboratory-based and the systems are very costly. In addition, the motion capture cameras can be affected by poor lighting conditions and often involve the use of a number of markers placed on the body, which is not practical for daily use.

Radar technology has appeared as the most suitable candidate for continuous gait monitoring at home due to its safety, simplicity, low cost, lack of contact, and unobtrusiveness while preserving privacy. Over the past decade, indoor gait measurement and analysis with different types of radars have been investigated and have showed promising results [13]. Most of the studies focused on the measurement of walking speed [14,15,16], while a few also attempted to measure instantaneous gait. Work [17] presented a study on the use of a 5.8 GHz pulse Doppler radar for gait speed and step time measurement. The feasibility of using radars to track different limb joints was first reported in [18] by using a continuous-wave (CW) radar working at 2.45 GHz, but no quantitative measure of gait parameters was presented. The most promising work on using radar for gait parameters extraction was reported in [19], where eleven biomechanical parameters were acquired using two 24 GHz CW radars. However, this work was carried out in a well-controlled environment, with the subjects walking on a treadmill.

In this work, we explored the use of millimeter waves for in-home gait analysis, and the intention was to continuously measure the gait parameters of clinical significance for the prediction and prevention of cognitive diseases and falls in the elderly. Several studies have reported that step (or stride) time variability (SVT) is a key gait feature in persons with neurodegenerative disorders, such as Parkinson’s disease and Alzheimer’s disease [20,21]. In addition, it has also been found that an increased STV was associated with fall risk. A one-year prospective study on fifty-two older adults suggested that stride time variability was significantly increased in those who subsequently fell compared with those who did not fall. It was also pointed out that although stride time variability correlated significantly with gait speed, the latter did not discriminate future fallers from nonfallers [22]. The same findings were verified by a study performed on persons with multiple sclerosis, and it was concluded that SVT may be a more sensitive marker of fall risk than the average walking speed [23]. SVT is usually expressed as a coefficient of variation, which is defined as the ratio of the standard deviation to the mean of the step (or stride) time [24]. Therefore, a reliable risk assessment relies on the accurate measurement of the step (or stride) time.

The aim of the study was to investigate the reliability of using a millimeter-wave frequency-modulated continuous wave (FMCW) radar for step time measurement in a real-life environment. There have been a few studies employing a millimeter-wave radar sensor for human recognition based on gait analysis [25,26,27,28], but, to the best knowledge of the authors, none of these studies have presented any quantitative measure of gait parameters.

## 2. Method

In this section, we describe the developed method for instantaneous gait analysis, including waveform design, measurement setup, data collection, and data analysis.

### 2.1. Hardware and Waveform Design

The millimeter-wave radar sensor used in this work was the AWR1642BOOST module from Texas Instruments [29]. The sensor operates from 77 GHz and supports a bandwidth of 4 GHz. The module includes two onboard-etched transmitting antennas and four receiving antennas. A data capture board DCA1000EVM [30] was used along with the radar sensor in order to capture the raw data.

As aforementioned, we were interested in capturing instantaneous gait features. Therefore, both good spatial resolution and time resolution are important. The achievable spatial resolution of the FMCW radar is determined by the chirp’s bandwidth, *B*, according to the following equation [31]:(1)$$\delta_{D}=\frac{c}{2B}=\frac{c}{2T_{c}\cdot S}$$
Here, *c* stands for the speed of light, $T_{c}$ is the chirp duration, and *S* is the chirp slope rate.

In the test, the bandwidth was chosen heuristically to be 3.6 GHz, which yields a 4.16 cm range resolution, giving a good possibility of tracking different limb joints. The chirp-to-chirp interval was set at 0.5 milliseconds, which allowed for a measurement of a maximum velocity of approximately 2 m/s according to the following relationship [31]:(2)$$\upsilon_{max}=\frac{\lambda }{4T_{s}}$$
Here, λ is the wavelength and $T_{s}$ is the chirp-to-chirp interval. We considered this velocity as high enough as our work is mainly intended for elderly care, where people usually walk very slowly.

Each frame was configured as one chirp and, in total, 30,000 frames were transmitted, which corresponds to a recording time of 15 s. The sampling rate needs to be high enough to resolve the frequency of the received signal, which is related to the distance of the subject according to the following equation [31]:(3)$$f_{r}=\frac{2S\cdot D}{c}$$
where *D* is the distance of the subject. We set the sampling rate at 6.25 MSa/s and the resulting maximum detection distance was approximately 5.86 m, which is a reasonable size for a room or a corridor in home environment.

The key radar parameters used in the work are given in Table 1.

### 2.2. Data Collection

The data collection was performed in two different environments as shown in Figure 1. One was a large and open gym hall that was mostly free of clutter, and the other was a lab corridor, which has a smaller free space and is considered to be a more challenging environment.

The measurements performed in the gym hall were mainly for finding an optimal measurement setup and developing a data analysis method. Different sensor heights and orientations were investigated in order to determine a suitable operating position, and, in the end, the sensor was placed at a height of 8 cm above the floor in order to focus the energy on the lower body parts, i.e., legs and feet, which are of the most interest for step time measurement. The subject was tasked with walking at a self-regulated pace along a 4 m long (from 5 m to 1 m away from the radar sensor) gym track with different types of gait, including away from or towards the radar sensor with a normal gait at different speeds, as well as imitated limp gaits. One of the limp gaits involved one leg swinging normally whereas the other sought support quickly after lift-off, and the other case was very similar but at a slower pace. During the measurements, video recordings were made by using a smartphone camera with the consent of the measured subject in order to provide the ground truth of some basic information, such as distance, number of steps, start and end time, etc. The track was marked at half-meter intervals for observational convenience, which made it convenient for the subject to step exactly at those intervals. This would result in exactly 8 steps, which gives a good reference for us to compare the radar measurement and video recordings.

A quantitative investigation on the step time measurement was performed in the lab corridor. Three healthy subjects (one female and two males) in the age range of 20–40 years were measured and each subject walked from approximately 6 m away towards the radar with a slow and self-regulated pace in 15 s. The same type of measurement was repeated ten times for each subject, resulting in a total measurement time of approximately 2.5 min each. During the measurements, the subjects wore two commercial motion tracking sensors (Xsens DOT) at the left and right ankle positions, respectively. The Xsens DOT sensor provides 3D angular velocity and acceleration using a gyroscope and an accelerometer [32]. The two motion sensors were synchronized during the measurement and the sampling rate of the motion sensor was 60 Hz. Best efforts were made to start the radar and motion sensor data collection at the same time.

The collected radar data were stored in a data matrix, as shown in Figure 2, corresponding to the transmitted frames. Each row represents the captured signal corresponding to each transmitted frame and is the “fast-time” data, and each column reflects the data change over frames and is the “slow-time” data.

### 2.3. Data Analysis

Figure 3a is the flow chart of the developed method for processing the collected data to obtain temporal gait parameters such as cadence and step time. Firstly, a fast Fourier transform (FFT) was applied to each row of the data matrix, as shown in Figure 2b, to generate a so-called “range profile”, which presents the locations of strongly reflected objects. By subtracting the range profile obtained from the first frame from others, peaks that correspond to static objects were eliminated and the remaining highest peaks were considered to be the measured subject. These peaks were then combined in time to form a range–time plot, which indicates the location of the measured subject over time.

A sliding window method (SWM) was then used to generate the gait pattern of the measured subject, and the main idea is illustrated in Figure 3b. With this method, the complete recording window was divided into many small frame windows (e.g., window 1…Nf) and, for each frame window, only a few range bins (one range bin corresponds to one column of the data matrix in Figure 2b) were selected for generating the gait pattern by referring to the range–time plot. The first step was to apply a short-time Fourier transform (STFT) to each selected range bin according to the following equation [33]:(4)$$S\left(n,k\right)=|\sum_{m=0}^{M-1}s\left(n+m\right)w\left(m\right)e^{-j2\pi mk/K}|$$
Here, *S* is the time–frequency spectrum, which is a two-dimensional representation of energy versus time and frequency. *s* is the slow-time signal for the selected range bin, which is the data along one column after the aforementioned FFT processing, and *w* is the window function. n=0,1,…,N−1 is the sample index of the signal, k=0,1,…,K−1 is the frequency index, and m=0,1,…,M−1 is the sample index of the window.

The time–frequency spectra generated for each range bin were added up and the resulting spectrum was rescaled to have a maximum value of one. This was carried out in order to diminish the effect of distance propagation loss. The rescaled spectrum was then thresholded to eliminate the background noise and the threshold value was determined by the following equation:(5)$$Threshold=\frac{\sum_{i=1,j=1}^{I,J}S_{b}\left(i,j\right)}{\sum_{i=1,j=1}^{I,J}S_{m}\left(i,j\right)}$$
Here, $S_{b}$ and $S_{m}$ are the time–frequency spectra obtained from measurements, where there is no subject and one subject walking, respectively. *I* and *J* define the size of the spectrum in the time and frequency dimensions.

After the thresholding, the spectrum was digitized to “zeros” and “ones”, resulting in a binary image. Connected components within a size of 30,000 pixels were treated as noise and thus taken away. The same process was applied to all of the frame windows and, at the end, the time–frequency spectra obtained for each frame window were aligned in time, resulting in a micro-Doppler signature for the entire recording window. The generated binary images for each frame window were also aligned, forming a complete gait pattern image.

From the micro-Doppler signature, we can calculate the cadence velocity diagram (CVD), which represents the repetition frequency of certain Doppler shifts:(6)$$\mathrm{CVD}\left(l,k\right)=|\sum_{n=0}^{N-1}S_{doppler}\left(n,k\right)e^{-j2\pi nl/L}|$$
Here, l=0,1,…,L−1 is the index of the Doppler repetition frequency. CVD enables a better analysis of the periodic patterns that are inherent to a human gait, such as the cadence, which is the number of steps per second.

The step time was estimated from the gait pattern image by extracting the upper envelope, which gives a good representation of the foot velocity over time.

This proposed method largely saved computational resources and time as it only needs to handle a small portion of the recorded data. Another advantage of this method is that it can give a high signal-to-noise ratio and robust handling of unknown measurement scenarios as it only deals with the location of interest under a specific time.

## 3. Results

This section includes important results obtained from the tests performed both in the gym hall and the lab corridor. Both qualitative and quantitative results are presented. Results from simultaneous measurements by using the millimeter-wave radar and a CW radar are also presented.

### 3.1. Micro-Doppler Signature and Cadence

Figure 4a,b are two measurement examples carried out in the gym hall, where the subject walked toward the radar sensor at a relatively fast and slow speed, respectively. Figure 4c,d are two similar measurements, but the subject was walking away from the radar sensor. The top plots are range–time plots, showing the distance between the subject and the radar sensor over time, and the middle and lower plots are the micro-Doppler signatures and the extracted envelope obtained by using the sliding window method shown in Figure 3.

The average walking speeds calculated from the range–time plots for the four measurement examples are approximately 0.48 m/s, 0.28 m/s, 0.37 m/s, and 0.30 m/s, respectively. These values are very close to those obtained from video recordings, which are 0.49 m/s, 0.29 m/s, 0.39 m/s, and 0.31 m/s.

Figure 4e,f shows the obtained micro-Doppler signature and the extracted envelope for the two test cases when the subject was asked to alter their gait in order to replicate a gait similar to a limp.

The cadence velocity diagram obtained according to Equation (Equation 6) and the mean cadence spectrum (mCS), which is computed as the average energy of each Doppler repetition frequency in the CVD, are shown in Figure 5 for the six measurement examples described above. The peaks in the mCS correspond to the harmonic components of the gait, where the peak close to DC corresponds to the oscillating movement of the torso whereas the second highest peak in the mCS tends to correspond to the oscillating motion of the legs in the case of an unassisted gait.

As the radar sensor’s radiation energy is mainly directed to the lower limb joints, the peaks corresponding to the leg/foot movement are the most prominent and are referred to as cadence. According to the mCS, the cadence for the first four test cases is 60 steps/min, 34 steps/min, 45 steps/min, and 35 steps/min, respectively. These values are in good agreement with the extracted envelope given in Figure 4 and the video recordings. However, the agreement does not hold for the limping cases. The cadence according to the mCS is 25 steps/min for both cases, which is significantly lower than the actual number of steps.

### 3.2. Quantitative Measure of Step Time

Figure 6a shows the range–time plot, micro-Doppler signature, and the extracted envelope of one test case performed in the lab corridor. The subject was out of the measurement zone at the beginning of the data recording and became “visible” at approximately 3.5 s. The differential of the extracted foot velocity in comparison with the acceleration data obtained with the Xsens DOT sensor is shown in Figure 6b. To provide a good illustration, the radar acceleration data were normalized and the Xsens sensor data were scaled and offset along the vertical axis. It can clearly be seen that every second peak of the radar acceleration data align well with the Xsens acceleration data recorded at the left ankle and right ankle, respectively. The time instants corresponding to these acceleration peaks were then used to calculate the step times.

The step times obtained from the radar data versus those obtained from the Xsens sensor data for all of the measurements are presented in Figure 7a. The calculated intra-class correlation coefficient (ICC) is 0.93. Figure 7b shows the relative error for each measurement subject, which is defined as the following equation:(7)$$\Delta =\frac{|ST_{radar}-ST_{Xsens}|}{ST_{Xsens}}\times 100\%$$
Here, $ST_{radar}$ and $ST_{Xsens}$ are the step time obtained from radar sensor and Xsens sensors, respectively.

The bottom whisker of each box indicates the lowest error and the topmost whisker indicates the highest one. The top and bottom edges of the blue rectangle indicate the 75th and 25th percentiles of the error, respectively. The red mark within the rectangle indicates the median of the error for each subject. Table 2 is a comparison of the step time measured by the radar and the Xsens sensors.

### 3.3. Comparison between the Millimeter-Wave FMCW Radar and a Microwave CW Radar

We conducted several simultaneous tests in the gym hall using the millimeter-wave radar and a microwave radar sensor. Figure 8a shows the measurement setup for the simultaneous measurement. The CW radar system is a software-defined ratio board (USRP2901 from National Instruments) operating from 70 MHz to 6 GHz [34]. A pair of wide-band bow tie antennas that provide conical coverage were used for transmitting and receiving signals [35]. The operating frequency was set at 2.45 GHz, which is the same as in the work [18]. The micro-Doppler signatures from two measurement examples are shown in Figure 8b–e. In the first case, the subject walked slowly along the defined track away from the sensors at a certain pace and, in the second case, the subject walked toward the sensors from five meters away with a corner reflector attached to one of the subject’s knees in order to enhance the corresponding echoes.

## 4. Discussion

In this section, we provide an explanation and interpretation of the presented results and compare our study with other works, as well as discuss the limitations and suggest future work.

### 4.1. Comments on the Results

For all of the measurement examples presented in Figure 4, the range–time plots are in good agreement with the video recordings in terms of the walking distance and walking time. The micro-Doppler signatures show clear patterns of each step, which facilitates envelope extraction. The extracted upper envelope shows the foot velocity as the subject walked, and we can clearly distinguish where each step ends and another begins. In general, we see a relatively lower foot top velocity when the subject was farthest away from the radar sensor, and this is highly attributed to the weaker reflections.

A big variation in step time is clearly seen in Figure 4e,f, where the subject was imitating a limp. The extracted envelope in Figure 4f is capable of capturing the limp near-perfectly as the stiff leg is presented with a much lower top velocity. These results demonstrate the good robustness of the developed method for detecting different types of gaits.

Whereas the cadence of the normal gait cases were correctly measured, we see an inaccurate measurement of the cadence for the altered gait cases. This suggests that cadence is a good measure of the pace of normal gaits, but not effective for abnormal gait cases.

Although the range–time plot shown in Figure 6a is more noisy than those shown in Figure 4, the quality of the obtained micro-Doppler signature is equally as high as those carried out in the gym hall. This indicates that the developed method is capable of dealing with a more challenging measurement environment, such as the lab corridor. Figure 7b shows that the measurement accuracy of the step time for the two males (subject B and C) is slightly better than that for the female (subject A), which may be attributed to the weaker reflections due to a smaller radar cross-section area of the female’s feet. We noticed that, in a few measurement tests for subject B, the acceleration peaks in the Xsens sensor data are not always prominent, which makes it challenging to identify the right peaks. This may be the reason for several outliers in the relative error plot. Overall, the median accuracy for the three subjects are all over 95%. The ICC of 0.93 indicates the excellent reliability of the proposed method for step time measurement.

The comparison test between the millimeter-wave radar and the 2.45 GHz CW radar clearly demonstrated the benefit of using millimeter waves for gait analysis. Especially for the second measurement example, where the subject wore the reflector on the knee while walking, the velocity pattern of the knee is largely enhanced and clearly seen in the micro-Doppler signatures obtained with the FMCW radar while the knee is hardly distinguishable from other body parts in CW measurements. This indicates that, with sufficiently strong radiation in a certain direction, we are able to track different limb joints using the millimeter-wave FMCW radar sensor. However, this is difficult to achieve by using the microwave CW sensor due to the insufficient spatial resolution.

The obtained results suggest that the millimeter-wave FMCW radar sensor is highly capable of capturing instantaneous gait features thanks to its inherent high spatial resolution. High accurate measurements of the step time were achieved with the developed method. The micro-Doppler signatures obtained with the millimeter-wave radar reveal much greater details of the gait pattern than those by using the microwave CW radar, which suggests a good potential for identifying different phases of a gait cycle and, as a consequence, giving a reliable measurement of more gait parameters.

### 4.2. Comparison with Other Works

As the proposed method is dedicated to in-home gait monitoring, where the use of wearable sensors is not preferable, the comparison to other works is limited to non-wearable methods. Paper [36] presented an electrostatic-sensing-based method where temporal gait parameters were estimated by sensing and analyzing the electrostatic field generated from human foot stepping. The reported average accuracy in the gait cycle measurement was 97% in the range of 3 m. Our method achieved almost the same measurement accuracy (96% in mean) in a range that was twice as long.

Work [17] reported an excellent reliability in the step time measurement in a lab environment by using a 5.8 GHz pulse Doppler radar with an ICC of 0.97 between the radar and a motion capture system. The drawback of the method is the relatively large size of the radar sensor and the use of a large metal shield (60 cm × 60 cm) at the back of the radar, which is unfavorable for integrating the system at home. In addition, two types of calibration need to be performed prior to the data collection.

Our work presents a simple method that needs neither calibration nor empirical values. Besides the available high spatial resolution, millimeter-wave radar has several other unique features that make it especially suitable for in-home gait analysis. For example, at the millimeter-wave range, the antenna size is very small, which allows for the deployment of an antenna array, resulting in a finer beam width. With effective beam-forming, most of the radiated energy can be directed to specific body parts, which consequently enhances the echoes of interest and reduces interference. Moreover, the millimeter-wave radar sensor is very compact and cheap, and can be easily integrated into furnitures and walls in a home environment.

### 4.3. Limitation of the Work

The presented work has several limitations. Firstly, the accuracy assessment of the step time measurement is subject to some errors. The sampling rate of the Xsens sensor is 60 Hz, providing a time resolution of 0.0167 s. In some of the measurements, the calculated ground truth may be off by one or two time instants compared to reality due to the difficulties in identifying the right peaks in the Xsens acceleration data, which correspond to a potential offset of ±0.0167 s to ±0.0333 s for each step.

Secondly, the data collection was limited to a small number of subjects. In order to make the collected data more representative, we included both male and female subjects in the study and mainly focused on a slow speed/walking pace. The step time of the collected data spans from approximately 0.9 s to 1.8 s, which represents a wide variation range. The next step is to perform a large-scale study among the target group (i.e., old adults) to further verify the developed method.

Thirdly, the developed method is highly dependent on the accurate localization of the measured subject, and this work demonstrated the effective tracking of a single subject in real-life environment with a single sensor. For the reliable tracking of multiple subjects and (or) a single subject in a more complex environment, e.g., including many pieces of furniture, the use of multiple sensors is preferred. The inherent tracking function of the employed FMCW radar makes it preferable for in-home monitoring in comparison to a CW radar, as it is less vulnerable to surrounding interference.

Lastly, although the FMCW radar sensor is configured with several antennas, only one antenna pair was used in this work for the sake of simplicity. It would be interesting to explore the use of multiple antenna pairs for beamforming to enhance echoes from the body parts of interest. As the focus of the work is to measure the step time, a low sensor elevation was chosen to focus the radiation energy on the foot in consideration of the antenna radiation pattern. In order to effectively capture the echoes from upper body parts, e.g., the torso, the deployment of a few sensors at different elevations is suggested.

## 5. Conclusions

The presented work demonstrated the reliable measurement of temporal gait parameters, and, more specifically, the step time, in real-life environments by using a millimeter-wave FMCW radar. These promising results suggest that this type of radar has good potential for the timely detection of discrete and subtle gait changes that appear at the early stage of cognitive disorders at home, enabling the prediction and prevention of cognitive diseases and fall accidents for the elderly.

## Figures and Tables

**Figure 1 sensors-22-09901-f001:**
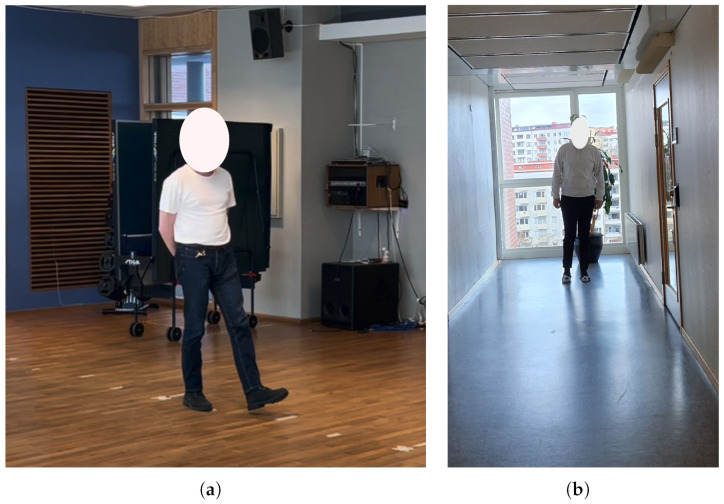
The two measurement environments for data collection: (**a**) a gym hall and (**b**) a lab corridor.

**Figure 2 sensors-22-09901-f002:**
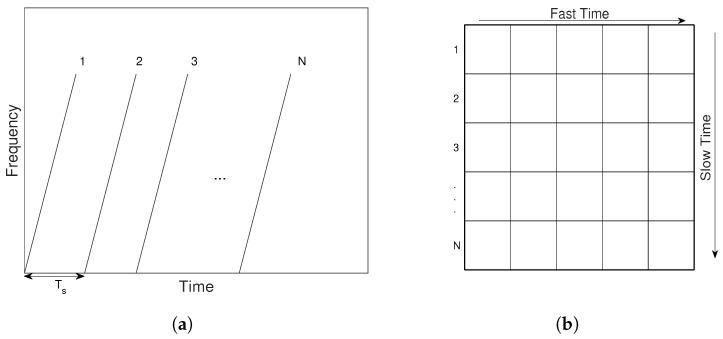
Recorded data corresponding to transmitted signals: (**a**) the transmitted frames and (**b**) the data matrix for storing the collected data.

**Figure 3 sensors-22-09901-f003:**
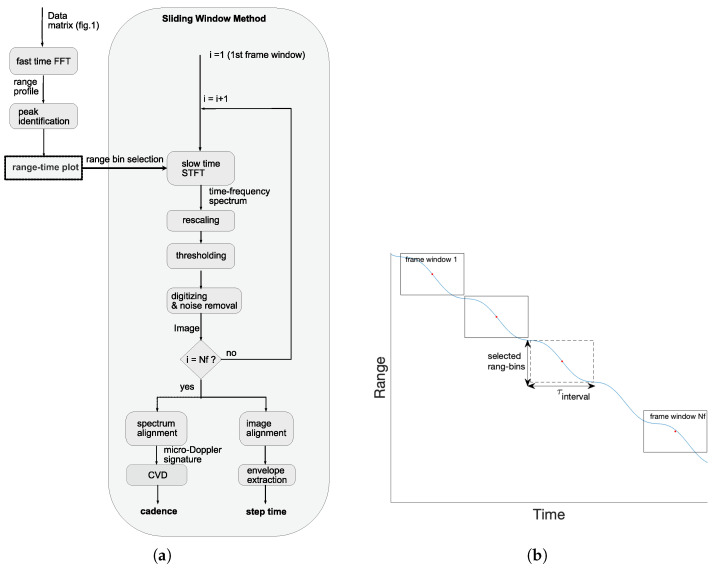
The data analysis method: (**a**) flow chart and (**b**) the illustration of the main idea of the sliding window method.

**Figure 4 sensors-22-09901-f004:**
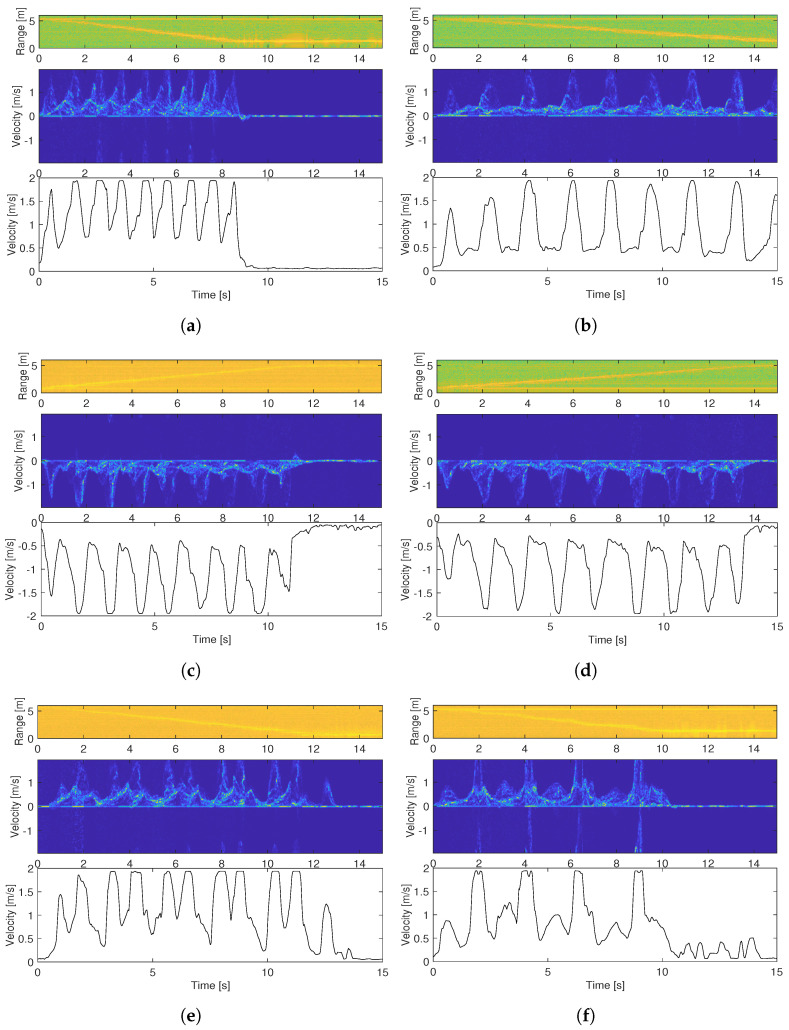
The range–time plot micro-Doppler signature, and extracted envelope for different measurement cases in the gym hall: (**a**,**b**) are two tests where the subject walked toward the radar sensor at different speeds; (**c**,**d**) are two tests where the subject walked away from the radar sensor at different speeds; (**e**,**f**) are two tests where the subject tried to imitate a limp with different intensities.

**Figure 5 sensors-22-09901-f005:**
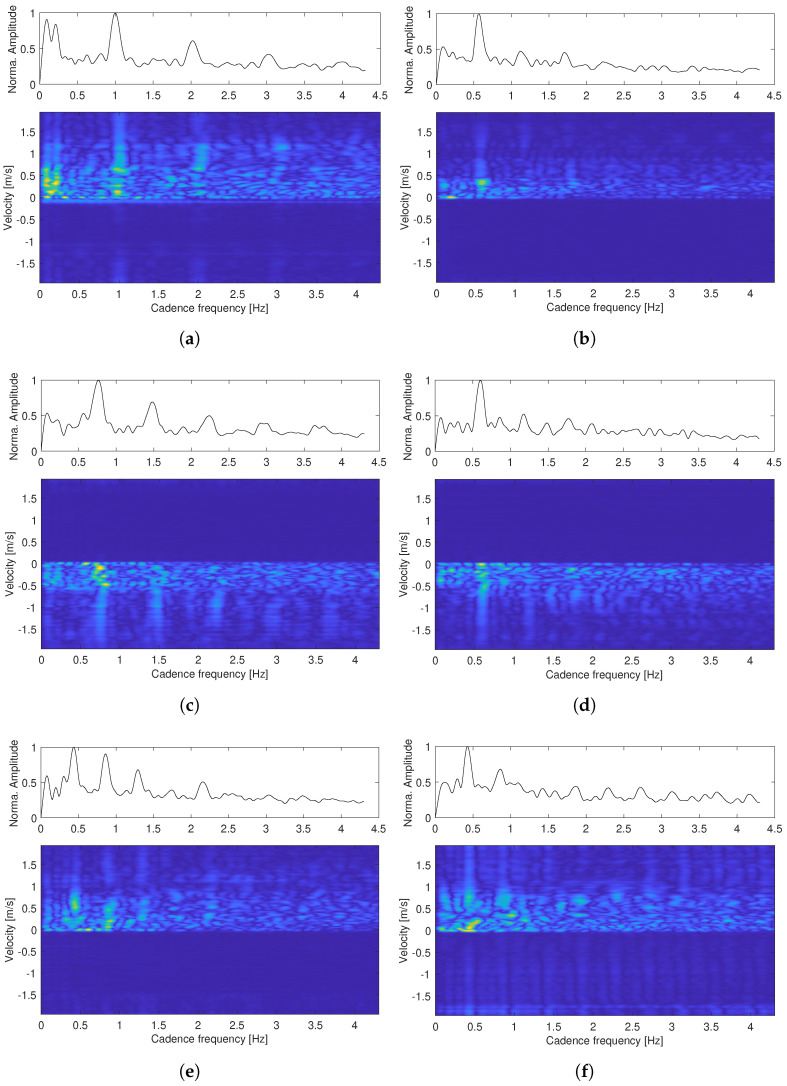
The cadence velocity diagram (lower plots) and mean cadence spectrum (upper plots) for the test cases (**a**–**f**) shown in Figure 4.

**Figure 6 sensors-22-09901-f006:**
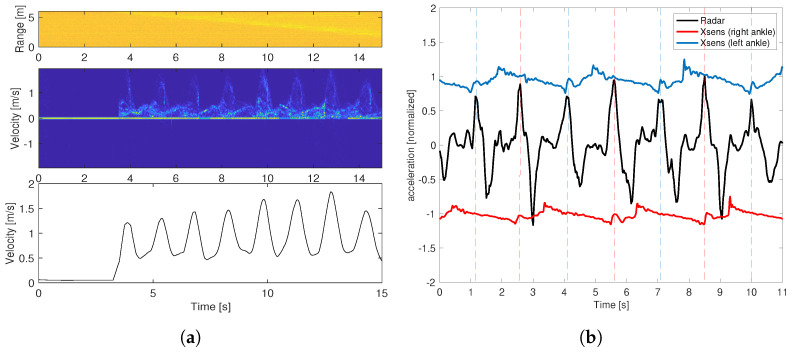
One measurement example for illustrating the calculation of the step time: (**a**) the range–time plot, micro-Doppler signature, and extracted envelope for one measurement example in the lab corridor; (**b**) comparison between acceleration data obtained with the radar and the Xsens sensors.

**Figure 7 sensors-22-09901-f007:**
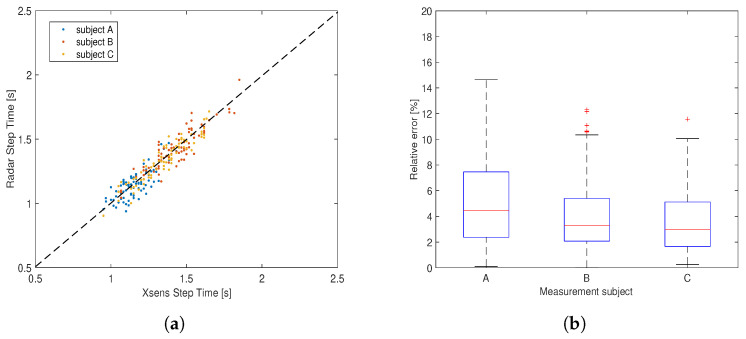
The measurement of step time by using the millimeter wave FMCW radar: (**a**) the plot of the step times obtained with the radar versus those obtained by using the Xsens sensor; (**b**) the relative error of the measured step time for each subject.

**Figure 8 sensors-22-09901-f008:**
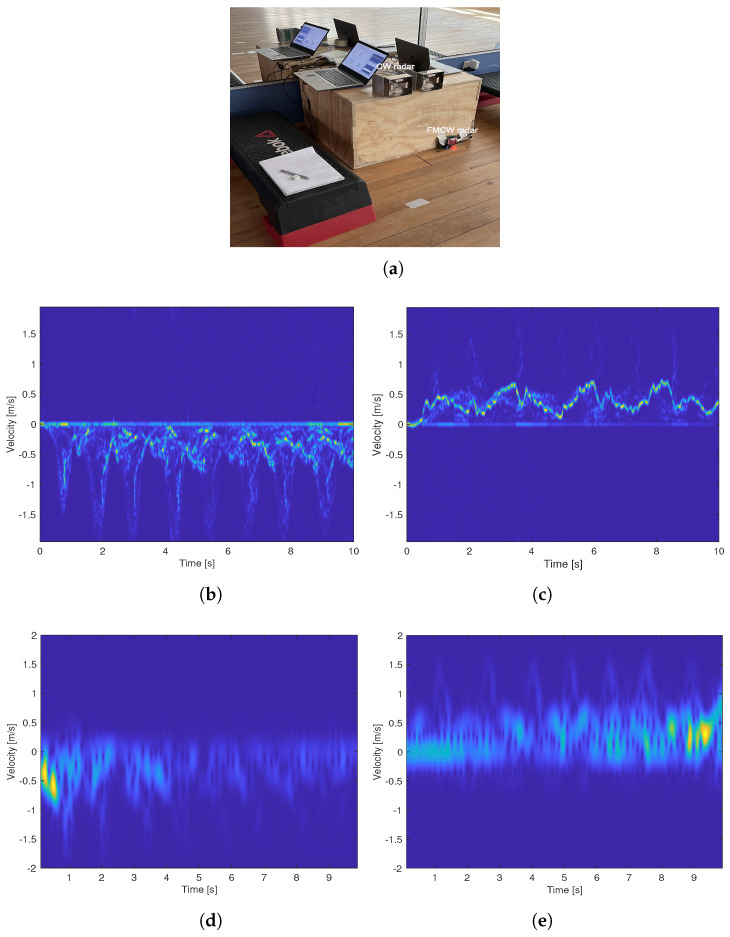
The comparison test between the millimeter-wave radar and a 2.45 GHz CW radar: (**a**) measurement setup; (**b**,**c**) are micro-Doppler signatures obtained by using the millimeter-wave radar for test case 1 and case 2, respectively; (**d**,**e**) are corresponding micro-Doppler signatures obtained with the CW radar.

**Table 1 sensors-22-09901-t001:** Key radar parameters used in the work.

Parameters	Value (unit)
Chirp slope rate	80 (MHz/μs)
Chirp duration	45 (μs)
Sampling rate	6250 (kSa/s)
Start frequency	77 (GHz)
No. chirps per frame	1
Frame-to-frame interval	0.5 (ms)
No. frames	30,000

**Table 2 sensors-22-09901-t002:** Comparison of step time measured by the radar and the Xsens sensors.

Subject	${\mathrm{ST}}_{\mathrm{Xsens}}$ (mean)	${\mathrm{ST}}_{\mathrm{radar}}$ (mean)	${\mathrm{ST}}_{\mathrm{Xsens}}$ (std)	${\mathrm{ST}}_{\mathrm{radar}}$ (std)	Δ (mean)	Δ (median)
A	1.1548	1.1484	0.1010	0.1112	0.0517	0.0443
B	1.4317	1.4249	0.1593	0.1583	0.0407	0.0331
C	1.3459	1.3384	0.1626	0.1640	0.0361	0.0296
A+B+C					0.0428	0.0362

## Data Availability

Data available on request from the authors.

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
