# Peer review of "Walking Step Monitoring with a Millimeter-Wave Radar in Real-Life Environment for Disease and Fall Prevention for the Elderly"

_sensors, 2022, doi:10.3390/s22249901_

Round 1
Reviewer 1 Report
The paper focuses on using millimeter-wave radar for gait analysis in a real-life environment. The authors tried to prove that their methodology performs better than continuous wave radar at 2.45GHz. The authors are focusing on walking step variability analysis of normal and abnormal gait, which is not reflected in the paper correctly. Hence the paper requires complete re-writing. Following are the comments on the paper,
1. Page 1, line 7, what is meant by the superior performance of the proposed design? What is detailed gait tracking?
2. Line 83, the authors claim that “Our work focused on the measurement of spatial-temporal gait parameters of clinical significance for disease and fall prediction.” In the introduction section, it is important to describe the gait parameters of clinical significance in more detail. Describe how the walking gait cycle and its different parameters are linked to the clinical evaluation.
3. Line 103, page 4, Please describe more about slow-time Fourier transform or provide a reference.
4. Line 146, page 5, What are the different types of gait? Describe more.
5. Page 6, how does figure 2 explain the extraction of different gait parameters?
6. Page 6, Line 177, a figure is required to explain how the microwave doppler signature and digital image combined together to show the gait pattern.
7. Page 7, Line 170, How Spatio-temporal gait parameters are calculated from the gait velocity alone? And what are those parameters?
8. Section 4.3, page 11, step time variability is a separate field of research that is not linked with gait parameters or patterns. However, if the authors are focusing on walking step variability analysis of normal and abnormal gait, then it should be clearly defined in the introduction, related work, and methodology.
9. A comprehensive analysis of different subjects is required to establish the efficacy of the proposed methodology.
10. Results should be compared with other methods published in the literature.
Author Response
We would like to thank the reviewer for valuable and constructive comments, which has helped us a lot for the improvement of the work. Please find our answers to each comment in the attached PDF document.

Reviewer 2 Report
Thank you very much for the opportunity to review the proposed manuscript. The investigators aimed to assess using a millimeter-wave frequency modulated
continuous wave radar for gait analysis in real life environment with a focus on the measurement of spatial-temporal gait parameters. The topic is interesting and important especially for elderly people; however, I have a few critical concerns with the proposed work.
1. Introduction is too long. Line 72-80 could be moved to Discussion section, and line 86-91 deleted. The aim of the study is not clearly stated.
2. The section 2 (Theory) could be omitted. If there is a need to use some equations in next sections, the Authors can use the equations there.
3. Methods unit is not well written. There is a lack of information about the patient, number of trials, pace? Did the same subject walk normally and replicate the abnormal gait? The description of the alter gait should be in a Method section as well as which parameters will be analyzed. Did the reference temporal parameters were taken from video recordings? Where were the cameras placed? What was the system?
4. Results needs revision. In this section only the data/results obtained from the study should be included here, not their explanation, interpretation, these parts should be moved to Discussion.
5. Did the Authors measure spatio-temporal parameters or only temporal gait parameters?
6. What are the limitations?
Author Response
We would like to thank the reviewer for the valuable and constructive comments, which has helped us a lot for the improvement of the work. Please find our answers to each comment in the attached PDF document.

Round 2
Reviewer 1 Report
I have checked the revised paper, and the authors have answered my comments in detail. I am satisfied with their answers. Only I have one comment. They should change the paper's title. Gait Analysis has a comprehensive meaning when we consider biomechanics evaluation. One possible suggestion is "Walking Step monitoring with a MillimeterWave Radar in Real-Life Environment towards Disease and Fall Prevention for the Elderly"
Author Response
We have now changed the title to "Walking Step monitoring with a MillimeterWave Radar in Real-Life Environment towards Disease and Fall Prevention for the Elderly"